# Sex-specific Mendelian randomization study of genetically predicted insulin and cardiovascular events in the UK Biobank

Jie V. Zhao[1], Shan Luo[1] & C. Mary Schooling [1,2]

Insulin drives growth and reproduction which trade-off against longevity. Genetically predicted insulin, i.e., insulin proxied by genetic variants, is positively associated with ischemic heart disease, but sex differences are unclear, despite different disease rates and reproductive strategies by sex. We used Mendelian randomization in 392,010 white British from the UK Biobank to assess the sex-specific role of genetically predicted insulin in myocardial infarction (MI) (14,442 cases, 77% men), angina (21,939 cases, 65% men) and heart failure (5537 cases, 71% men). Genetically predicted insulin was associated with MI (odds ratio (OR) 4.27 per pmol/L higher insulin, 95% confidence interval (CI) 1.60 to 11.3) and angina (OR 2.93, 1.27 to 6.73) in men, but not women (MI OR 0.80, 95% CI 0.23 to 2.84, angina OR 1.10, 95% CI 0.38 to 3.18). Patterns were similar for insulin resistance and heart failure. Mitigating the effects of insulin might address sexual disparities in health.

---

[1] School of Public Health, Li Ka Shing Faculty of Medicine, The University of Hong Kong, Hong Kong SAR, China. [2] City University of New York, School of Public Health and Health Policy, New York, NY, USA. Correspondence and requests for materials should be addressed to C.M.S. (email: cms1@hku.hk)

Cardiovascular disease (CVD) is the leading cause of global morbidity and mortality[1], accounting for over 30% of all deaths[2]. This burden of disease calls for more effective prevention and treatment strategies. Notably, men have higher incident rates of ischemic CVD than women, for reasons which have not been fully explained by traditional cardiovascular risk factors, such as smoking, lipid profile and blood pressure[3]. Clarifying the sex disparity in CVD might provide clues to identifying new targets as well as addressing sexual disparities in health.

A novel explanation for the sex difference, from the perspective of evolutionary biology, is that longevity trades-off against growth and reproduction, with possibly sex-specific trade-offs for reproductive investment versus cardiovascular risk[4,5]. Genetic evidence consistent with ischemic heart disease (IHD) trading off against fertility exists[6]. Genetically predicted gonadotrophin releasing hormone increases IHD[7]. Applying this concept to disease prevention and treatment, factors that downregulate growth or reproduction might lower the burden of CVD[8], differentially in men and in women, as drivers of reproduction are sex specific. Insulin is a key driver of growth and reproduction[9,10], which is responsive to nutritional factors, such as a high-fat diet[11], and to treatment with insulin and insulin secretagogues[12].

Insulin is an invaluable life-saving treatment for type 1 diabetes[13]. Unexpectedly and controversially, use of insulin has long been suspected to play a role in CVD[14], especially in men[15]. Genetically predicted insulin and insulin resistance are consistently positively associated with higher risk of IHD[16–18], independent of adiposity[18]. Patients switching from metformin to an insulin secretagogue, sulphonylurea, have a higher risk of myocardial infarction (MI)[19]. Higher insulin doses are also associated with an unfavorable cardiovascular risk factor profile[20]. No large randomized controlled trials (RCTs) have assessed the role of insulin in CVD, and it may not be ethical to do so. In these circumstances, Mendelian randomization (MR) taking advantage of genetic endowment randomly allocated at conception[21], can obtain unconfounded estimates. Here, we used MR to assess overall and sex-specific effects of insulin, and for completeness insulin resistance, on MI, angina, heart failure and their key risk factors (low-density lipoprotein (LDL) cholesterol, apolipoprotein B (ApoB)[22], and blood pressure) using individual data in a large cohort, the UK Biobank[23], or the largest available genome-wide association study (GWAS). Red blood cell attributes have long been suspected to be relevant to cardiovascular disease[24], however, which trait matters is not well established. The most recent evidence from an MR study suggests the red blood cell trait, reticulocyte count, may be a causal factor for IHD[25], so we similarly examined the role of insulin and insulin resistance in reticulocyte count. We also validated the findings for MI using summary statistics from a large genetic study, CARDIoGRAMplusC4D 1000 Genomes[26].

## Results

**Genetic instruments for insulin and insulin resistance**. We identified 12 single-nucleotide polymorphisms (SNPs) independently predicting insulin and 12 SNPs for BMI-adjusted insulin (extracted from Table 1 of the GWAS of Scott et al.[27]), as previously[17,28]. All reached genome-wide significance ($5 \times 10^{-8}$). The insulin resistance genetic score was constructed based on 10 SNPs (rs4846565, rs10195252, rs2943645, rs17036328, rs3822072, rs6822892, rs4865796, rs459193, rs2745353 and rs731839), as in the previously validated score[29].

Of the 12 SNPs related to insulin, we dropped 5 SNPs due to pleiotropy, specifically 2 SNPs (rs10195252 in *GRB14* and rs9884482 in *TET2*) related to alcohol drinking (p value $9.4 \times 10^{-7}$ and $2.3 \times 10^{-6}$, respectively), 2 SNPs (rs1167800 in *HIP1* and rs7903146 in *TCF7L2*) related to BMI (p value $8.7 \times 10^{-13}$ and $9.8 \times 10^{-7}$, respectively), and 1 SNP (rs1421085) in the *FTO* gene (a well-established obesity predictor), so 7 SNPs were used (Table 1 and Supplementary Table 1).

Of the 12 SNPs related to insulin adjusted for BMI, we dropped 3 pleiotropic SNPs, specifically, 2 SNPs (rs974801 in *TET2* and rs10195252 in *GRB14*) related to alcohol drinking (p value $8.5 \times 10^{-7}$ and $9.4 \times 10^{-7}$, respectively) and 1 SNP (rs6912327 in *C6orf107*) related to BMI (p value $= 1.6 \times 10^{-14}$) in the UK Biobank, leaving 9 SNPs (Table 1).

In sensitivity analysis for insulin resistance in men, we further dropped 1 SNP (rs3822072 in *FAM13A1*) associated with BMI in men (p value $= 5.5 \times 10^{-6}$) but not women. Given the unclear causal role of alcohol in CVD, we kept the SNPs related to alcohol drinking in sensitivity analysis. For completeness, we also kept the SNPs related to BMI (Table 1).

**Associations with MI, angina and heart failure**. Genetically predicted insulin, BMI-adjusted insulin and insulin resistance score were all positively associated with MI overall (Table 2 and Supplementary Fig. 1) and also in men, but not in women (p values for sex differences were 0.02, 0.04 and 0.04 respectively) (Table 2). The overall association was validated in CARDIoGRAPMplusC4D 1000 Genomes (Table 2). Insulin and insulin resistance were also associated with higher risk of angina in men only (Table 2), with a more obvious sex difference for BMI-adjusted insulin (p value for sex difference 0.04), than insulin (p value for sex difference 0.08) (Table 2). The pattern of associations were generally robust to different analytic methods (Supplementary Table 2), and sensitivity analysis including alcohol-related SNPs (Supplementary Table 3), all potentially pleiotropic SNPs (Supplementary Table 4) and excluding rs3822072 for BMI-adjusted insulin in men (Supplementary Table 5), despite some associations including the null value due to wider confidence intervals.

The replication for MI using a different study provides additional validation, and enabled us to test causality in a cost-efficient way[30]. Specifically, the studies for MI with over 56,000 cases, at an approximate $R^2$ of 0.01 (variance in insulin/BMI-adjusted insulin explained by the genetic predictors), has 0.8 power to detect an odds ratio (OR) of about 1.13 per one standard deviation increase in the exposure. The UK biobank has 0.8 power to detect an OR of about 1.27 for MI overall, of 1.31 for MI in men and 1.64 in women; an OR of 1.22 for angina overall, of 1.28 for angina in men and 1.39 in women; an OR of 1.46 for heart failure overall, of 1.57 for heart failure in men and 2.01 in women[31]. The larger number of cases in men than women enabled us to test a smaller effect size in men, however, there is sufficient power for both men and women when using the insulin resistance score on angina. The difference in power does not explain the sex-disparity in the magnitude of the point estimates and/or direction of associations.

**Associations with CVD risk factors**. Insulin and BMI-adjusted insulin were unrelated to LDL cholesterol, but the latter was associated with higher ApoB (Fig. 1). We could not test whether these associations differ by sex, because relevant genetic data is not publicly available. Insulin and BMI-adjusted insulin were positively associated with systolic blood pressure and reticulocyte count in both men and women. The associations of BMI-adjusted insulin with reticulocyte count appeared to be stronger in men than in women, although the sex difference was not statistically significance (p value for sex-difference 0.17).

**Table 1 Genetic predictors for insulin and insulin adjusted for BMI**

| Exposures | SNPs | Gene | Effect allele | Beta | *p* Value | Note |
|---|---|---|---|---|---|---|
| Insulin | rs1530559 | *YSK4* | A | 0.015 | $3.4 \times 10^{-8}$ | |
| | rs2745353 | *RSPO3* | T | 0.014 | $5.5 \times 10^{-9}$ | |
| | rs2820436 | *LYPLAL1* | C | 0.015 | $4.4 \times 10^{-9}$ | |
| | rs2972143 | *IRS1* | G | 0.014 | $3.2 \times 10^{-8}$ | |
| | rs4865796 | *ARL15* | A | 0.015 | $2.1 \times 10^{-8}$ | |
| | rs731839 | *PEPD* | G | 0.015 | $1.7 \times 10^{-8}$ | |
| | rs983309 | *PPP1R3B* | T | 0.029 | $3.8 \times 10^{-14}$ | |
| | rs1167800 | *HIP1* | A | 0.016 | $2.6 \times 10^{-9}$ | Dropped in main analysis due to potentially pleiotropic association with BMI; included in sensitivity analysis |
| | rs7903146 | *TCF7L2* | C | 0.018 | $6.1 \times 10^{-11}$ | |
| | rs1421085 | *FTO* | C | 0.020 | $1.9 \times 10^{-15}$ | |
| | rs9884482 | *TET2* | C | 0.017 | $1.4 \times 10^{-11}$ | Dropped in main analysis due to potentially pleiotropic association with alcohol; included in sensitivity analysis |
| | rs10195252 | *GRB14* | T | 0.016 | $4.9 \times 10^{-10}$ | |
| Insulin adjusted for BMI | rs17036328 | *PPARG* | T | 0.021 | $3.6 \times 10^{-12}$ | |
| | rs2126259 | *PPP1R3B* | T | 0.024 | $3.3 \times 10^{-13}$ | |
| | rs2943645 | *IRS1* | T | 0.019 | $2.3 \times 10^{-19}$ | |
| | rs3822072 | *FAM13A1* | A | 0.012 | $1.8 \times 10^{-8}$ | |
| | rs459193 | *ANKRD55* | G | 0.015 | $1.2 \times 10^{-10}$ | |
| | rs4846565 | *LYPLAL1* | G | 0.013 | $1.8 \times 10^{-9}$ | |
| | rs4865796 | *ARL15* | A | 0.015 | $2.2 \times 10^{-12}$ | |
| | rs6822892 | *PDGFC* | A | 0.014 | $2.6 \times 10^{-10}$ | |
| | rs731839 | *PEPD* | G | 0.015 | $5.1 \times 10^{-12}$ | |
| | rs6912327 | *C6orf107* | T | 0.017 | $2.3 \times 10^{-8}$ | Dropped in main analysis due to potentially pleiotropic association with BMI; included in sensitivity analysis |
| | rs974801 | *TET2* | G | 0.014 | $3.3 \times 10^{-11}$ | Dropped in main analysis due to potentially pleiotropic association with alcohol; included in sensitivity analysis |
| | rs10195252 | *GRB14* | T | 0.017 | $1.3 \times 10^{-16}$ | |

*BMI* body mass index

**Table 2 Associations of genetically predicted insulin, insulin adjusted for BMI and insulin resistance genetic score with myocardial infarction, angina, and heart failure**

| Outcomes | Data source | Sex | No. of cases | Insulin | | | Insulin adjusted for BMI | | | Insulin resistance genetic score | | |
|---|---|---|---|---|---|---|---|---|---|---|---|---|
| | | | | OR | 95% CI | *p* | OR | 95% CI | *p* | OR | 95% CI | *p* |
| Myocardial infarction | UK Biobank | Overall | 14,442 | **2.87** | **1.30 to 6.33** | **0.009** | **2.63** | **1.36 to 5.07** | **0.004** | **3.23** | **1.88 to 5.56** | **$2.4 \times 10^{-5}$** |
| | | Men | 11,182 | **4.27** | **1.60 to 11.3** | **0.004** | **3.60** | **1.46 to 8.83** | **0.005** | **4.17** | **2.24 to 7.77** | **$6.9 \times 10^{-6}$** |
| | | Women | 3260 | 0.80 | 0.23 to 2.84 | 0.73 | 0.96 | 0.30 to 3.09 | 0.95 | 1.41 | 0.46 to 4.29 | 0.55 |
| | CARDIoGRAMplusC4D 1000 Genomes | Overall | 42,561 | **1.90** | **1.04 to 3.49** | **0.04** | **2.20** | **1.08 to 4.50** | **0.03** | -- | -- | -- |
| | Meta-analysis | Overall | 57,003 | **2.21** | **1.37 to 3.58** | **0.001** | **2.42** | **1.49 to 3.93** | **0.0003** | -- | -- | -- |
| Angina | UK Biobank | Overall | 21,939 | 2.05 | 0.87 to 4.83 | 0.10 | 1.90 | 0.92 to 3.94 | 0.08 | **2.87** | **1.84 to 4.48** | **$3.7 \times 10^{-6}$** |
| | | Men | 14,331 | **2.93** | **1.27 to 6.73** | **0.01** | **2.74** | **1.21 to 6.19** | **0.02** | **3.56** | **2.04 to 6.22** | **$8.3 \times 10^{-6}$** |
| | | Women | 7608 | 1.10 | 0.38 to 3.18 | 0.87 | 1.00 | 0.46 to 2.19 | 1.00 | 1.96 | 0.93 to 4.12 | 0.08 |
| Heart failure | UK Biobank | Overall | 5537 | 0.98 | 0.37 to 2.61 | 0.97 | 1.18 | 0.45 to 3.14 | 0.74 | 1.00 | 0.42 to 2.35 | 1.00 |
| | | Men | 3935 | 1.17 | 0.30 to 4.64 | 0.82 | 1.90 | 0.62 to 5.82 | 0.26 | 1.71 | 0.62 to 4.74 | 0.30 |
| | | Women | 1602 | 0.64 | 0.06 to 7.18 | 0.72 | 0.37 | 0.03 to 4.05 | 0.41 | 0.28 | 0.06 to 1.34 | 0.11 |

*BMI* body mass index, *CI* confidence interval, *OR* odds ratio
Inverse variance weighting with random effects was used for insulin and insulin adjusted for BMI; logistic regression was used for insulin resistance genetic score
The bold values denote associations with confidence intervals not including the null

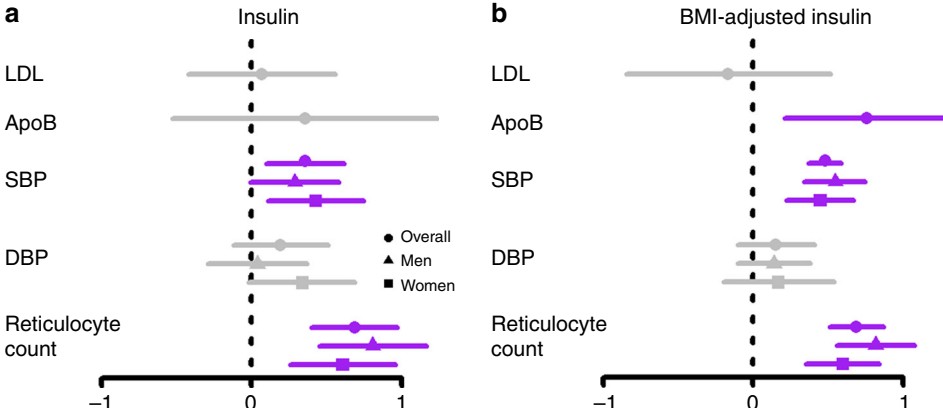

**Fig. 1** Associations of genetically predicted insulin and BMI-adjusted insulin with cardiovascular disease risk factors overall and by sex. ApoB apolipoprotein, BMI body mass index, DBP diastolic blood pressure, LDL low-density lipoprotein, SBP systolic blood pressure. Beta coefficients and 95% confidence intervals (CI) for the associations of insulin and BMI-adjusted insulin with CVD risk factors have been depicted. Gray denotes the 95% CI included the null, purple denotes the 95% CI did not include the null. $n = 188,577$ for LDL cholesterol, $n = 24,925$ for ApoB, $n \leq 361,194$ for blood pressure and reticulocyte count

## Discussion

Our Mendelian Randomization study suggests a positive association of insulin and insulin resistance with MI overall and in men, and with angina in men but not women, with validation for MI overall in CARDIoGRAPMplusC4D 1000 Genomes. Insulin and BMI-adjusted insulin increased blood pressure and reticulocyte count and BMI-adjusted insulin increased ApoB. We found no effect of insulin on LDL cholesterol, but sex-specific analysis for LDL cholesterol and ApoB were not examined thus sex differences cannot be excluded.

Our findings, together with previous MR studies, provide support for a potential role of insulin in IHD[14,32,33]. Our novel study also adds to the very limited evidence on the sex-specific effects of insulin and insulin resistance, by showing a stronger effect for men than women. Nevertheless, our study has limitations. First, MR is based on three stringent assumptions, i.e., the genetic variants are strongly related to the exposure, are not related to the exposure-outcome confounders, and the genetic variants are related to the outcomes only via influencing the exposure[34,35]. To satisfy the first assumption, we used genetic variants strongly associated with insulin and insulin resistance from a large GWAS[27,29], as previously[16,17]. To satisfy the second assumption, we checked for associations with known exposure-outcome confounders, including socioeconomic position and lifestyle in the UK Biobank, where there was no association with these potential confounders. In addition, the sample for genetic variants on insulin has no overlap with the UK Biobank. Two-sample MR is usually less biased than one-sample MR[36], because any relation of the genetic variants with unmeasured confounders is not expected to exist coincidently in both the sample providing genetic associations with insulin or insulin resistance and the sample providing genetic associations with the outcomes, due to the different data structures[37]. If bias did occur due to weak instruments, it is often towards the null, whereas in one-sample MR the bias is towards the direction of the conventional observational studies[36]. Population stratification might affect genetic distribution and cardiovascular risk, however, we only used participants of European ancestry, with genetic control. To test the assumption of pleiotropy, we checked for the known potential pleiotropy in three comprehensive curated databases. We also tested and corrected for pleiotropic effects using MR-PRESSO. Second, although we used the largest available source of genetic associations with heart failure, the number of cases was relatively

low, which may explain the wide confidence intervals for heart failure. Third, our MR study assessing the role of endogenous insulin and insulin resistance might not be applicable to the exogenous use of insulin. MR examining a lifetime effect of an exposure may also not be comparable to exogenous treatment. However, serum insulin is responsive to exogenous supplementation, diet, and treatment[12]. The association for MI is also consistent with the higher cardiovascular risk when switching to sulphonylurea, an insulin secretagogue[19]. Fourth, our study could be affected by selection bias from selecting survivors of their genetic make-up[38], and of competing risk of other specific causes of death that share risk factors. Specifically, the estimates for a potentially harmful exposure might be biased towards being less harmful if people with higher levels of exposures were already dead and not selected into the study, as in the obesity paradox[39]. Fifth, misclassification of the outcomes might exist. Measurement error in the outcomes might arise, but likely non-differential and so biases towards the null. Sixth, the associations in Europeans may not apply to other populations, such as Asians. However, causal effects should be consistent across settings, although their relevance may vary by population. Specifically, our findings may be particularly relevant to Asians who tend to have higher serum insulin than people of European descent[40]. Seventh, some of the participants may have comorbidities such as type 2 diabetes and may be taking medications for these comorbidities. Co-morbidities and their treatment may affect the cardiovascular outcomes, but should not affect the genetic predictors of exposures, so they are not confounders but their inclusion could improve the precision of the estimates. However, co-morbidities could also be consequences of insulin and insulin resistance so their consideration in the model would give the direct effects of insulin rather than the total effect sought, i.e., might create bias. As such, we did not account for co-morbidities or their treatment by adjustment or restriction, so as to obtain an unbiased, though possibly less precise, estimates. Eighth, reverse causality may occur if people with cardiovascular events change their lifestyle thereby affecting insulin or insulin resistance. However, these changes would not affect genetically predicted insulin or insulin resistance. None of the genetic variants are genome-wide significant ($p < 5 \times 10^{-8}$) for cardiovascular events, so it is unlikely that they predict insulin or insulin resistance by affecting cardiovascular events. Finally, the genetic variants in the sex-specific analysis were from both sexes rather than specifically for men and

women. As such, the sex-specific associations are less precise may be conservative. However, the directions of the associations should be unchanged. Validation of the sex-specific associations in another cohort is warranted.

Our study, together with previous evidence[41,42], suggests that insulin and insulin resistance have symbiotic roles that may both ultimately play a role in CVD. Our study adds to the current evidence by showing a sex-disparity in these associations. Insulin resistance has previously been proposed as a mechanism underlying the metabolic syndrome and hence susceptibility to CVD, i.e., Reaven's hypothesis[43]. However, before the advent of MR, Reaven's hypothesis was difficult to test conclusively, and included cholesterol, when ApoB, rather than LDL cholesterol, may be driven by insulin and correspond better to the target of lipid-modifying treatment[22]. Insulin modulates human ApoB mRNA translation[44]. Verification in RCTs would be worthwhile. Insulin might operate via increasing growth or sex hormones, such as androgens[45], which may increase the risk of MI[46]. Coagulation factors, which were not examined in this study, may also be a mechanism. Hyperinsulinemia promotes a procoagulant state[47], increasing in several coagulation factors, such as thrombin generation[47], and plasminogen activator inhibitor type 1[48], which have been identified as potential causes of IHD and MI[49,50].

Our study suggests that a lifestyle, which lowers serum insulin, might lower cardiovascular risk. Conversely, less limited living conditions that enable higher levels of insulin, with corresponding effects acting on MI and angina via male reproductive factors may explain the higher rates of CVD in men than women that emerge with economic development[51]. Insulin causing MI may also partly explain the unexpected off-target effects of insulin raising treatments, such as sulfonylureas[19]. Similarly, insulin therapy has a relatively higher risk of MI than insulin sensitization therapy[48]. Several medications for type 2 diabetes, such as metformin, thiazolidinediones, sodium-glucose transport inhibitors, and glucagon-like peptide 1 agonists may reduce the need for insulin[52]. From the perspective of clinical and public health practice, our findings suggest that medications or dietary factors that operate other than by increasing insulin might prevent and treat CVD. Our findings also draw attention to the possibility that insulin may operate via red blood cell attributes, such as reticulocyte count. Clarifying these pathways, especially as regards any sex-differences in the response to insulin, would be valuable, with relevance to the re-positioning of existing drugs and new drug development. Replication in other cohorts is needed.

## Methods
We used a two sample MR study design. Specifically, we obtained genetic predictors for insulin and insulin resistance from the largest available GWAS[27], and examined their overall and sex-specific associations with MI, angina and heart failure in the UK Biobank, and with LDL cholesterol, ApoB, blood pressure, and reticulocyte count in the largest available GWAS. For MI, we conducted a validation using CARDIoGRAMplusC4D 1000 Genomes. All the data sources were shown in Fig. 2.

**Genetic associations with insulin and insulin resistance**. The exposures were genetically predicted insulin and insulin resistance. As previous MR study on insulin resistance[28], we used insulin adjusted for body mass index (BMI) as an indicator of insulin resistance. We also validated the known insulin resistance findings using a validated genetic score for insulin resistance[29]. Genetic associations with all exposures were taken from a large meta-analysis of GWAS, conducted in adults ($n = 108,557$; mean age, 50.6 years; ~53% men) of European ancestry, without diabetes, adjusted for age, sex, study site and geographic covariates using an additive genetic model[27]. We did not use a genetic instrument for insulin resistance encompassing a lipid phenotype so as to focus more on insulin[53].

To meet the three key assumptions of instrumental variable analysis, i.e., relevance, independence and exclusions-restriction[34], we used genetic variants strongly and independently ($r^2 < 0.01$) predicting the exposures. We used LD-Link (https://ldlink.nci.nih.gov/) based on Europeans to check for correlations (linkage disequilibrium) between genetic variants. We checked whether these genetic variants were independent of potential confounders from their association with Townsend index, smoking, alcohol drinking, physical activity and BMI in the UK

Biobank. We dropped single nucleotide polymorphisms (SNPs) associated with any of these potential confounders at Bonferroni corrected significance ($p < 0.05/2000$ (number of phenotypes in the UK Biobank) = $2.5 \times 10^{-5}$). We checked for known direct effects of the exposures on the outcomes (violation of the exclusion-restriction assumption) in three comprehensive curated genotype to phenotype cross-references, i.e., Ensembl (http://www.ensembl.org/index.html), the GWAS catalog (https://www.ebi.ac.uk/gwas/) and PhenoScanner (www.phenoscanner.medschl.cam.ac.uk).

**Genetic associations with MI, angina and heart failure**. Genetic associations with MI, angina and heart failure were obtained using individual-level data in the UK Biobank (under application #42468), with validation for MI using summary statistics from CARDIoGRAPMplusC4D 1000 Genomes[26].

The UK Biobank is an ongoing large prospective cohort study[23]. The UK Biobank recruited 502,713 people aged 40–69 years, mean age 56.5 years, from Great Britain between 2006 and 2010, with 94% self-reported European ancestry, 45.6% men and median follow-up time currently 11.1 years. Disease outcomes were obtained from a nurse-led interview at recruitment, with ongoing follow-up via record linkage to all health service encounters and deaths[54]. Genotyping was assessed using two very similar arrays, i.e., the UK BiLEVE array and UK Biobank Axiom array. To control for population stratification, we restricted our analysis to participants with self-reported and genetically validated white British ancestry. For quality control, we also excluded participants with (1) excess relatedness (more than 10 putative third-degree relatives) or (2) mismatched information on sex between genotyping and self-report, or (3) sex-chromosomes not XX or XY, or (4) poor-quality genotyping based on heterozygosity and missing rates >1.5%. After quality control, we identified 392,010 white British in the UK Biobank, with 14,442 cases of MI (77% men), 21,939 cases of angina (65% men), and 5,537 cases of heart failure (71% men). Genetic associations with MI, angina and heart failure were obtained using logistic regression controlling for age, assay array and 10 principal components in sex-specific analysis and additionally adjusted for sex in the overall analysis, as the adjustment in our previous MR study in the UK Biobank[46].

Data on coronary artery disease/MI have been contributed by CARDIoGRAMplusC4D investigators and have been downloaded from www.cardiogramplusc4d.org. CARDIoGRAMplusC4D 1000 Genomes is a large genetic study (IHD cases $n = 60,801$, others = 123,504), with ~70% of the cases MI. The participants are mainly of European descent (77%) with phenotyping based on medical records, clinical diagnosis, as well as medications, or indicative symptoms or procedures, such as revascularization, and/or angiographic evidence of stenosis[26].

**Genetic association with CVD risk factors**

*Genetic association with LDL cholesterol and ApoB*. Genetic associations with LDL cholesterol (inverse normal transformed effect sizes), adjusted for age, age$^2$ and sex, were obtained from the Global Lipids Genetics Consortium Results summary statistics including 188,577 participants of European descent and 7,898 participants of non-European descent, mean age 55.2 years[55]. Genetic associations with ApoB (inverse normal transformed effect sizes), adjusted for age, sex, time from last meal, if applicable, and first ten principal components, were obtained from a meta-analysis GWAS of metabolomics in 24,925 Europeans (45% men)[56]. In both GWAS, genomic control was applied to each sample and the meta-analysis results, to correct for inflated test statistics due to potential population stratification.

*Genetic associations with blood pressure and reticulocyte count*. We obtained overall and sex-specific genetic associations with blood pressure and reticulocyte count using summary statistics from the UK Biobank, provided by Neale Lab (http://www.nealelab.is/uk-biobank/) in 361,194 white British (167,020, 46% men). The study adjusted for age, age$^2$, and 20 principal components in sex-specific analysis and additionally adjusted for sex and interactions of sex with age and age$^2$ in the overall analysis.

**Statistics and reproducibility**. We obtained MR estimates for the associations of genetically predicted insulin with MI, angina, heart failure and cardiovascular risk factors from two-sample instrumental variable analysis. Specifically, we obtained SNP-specific Wald estimates (quotient of genetic association on outcome and genetic association on insulin) and then meta-analyzed them using inverse variance weighting (IVW) with multiplicative random effects. We validated the findings for MI using summary statistics from a large genetic study, CARDIoGRAMplusC4D 1000 Genomes (IHD cases $n = 60,801$, others = 123,504), with ~70% of the cases MI[26]. A consistent direction in both studies gives more confidence. As sex-specific summary statistics were not available in CARDIoGRAMplusC4D 1000 Genomes, we only used overall statistics to replicate overall associations with MI. For insulin resistance genetic score, logistic regression was used to obtain the associations of the genetic score with MI, angina and heart failure controlling for age, assay array and 10 principal components in sex-specific analysis and additionally adjusted for sex in the overall analysis, as previously[46]. Power calculations were conducted overall and by sex. MR studies require larger sample sizes than conventional observational studies, because the sample size needed for MR is the sample size for the conventional observational study divided by the variance in the exposure

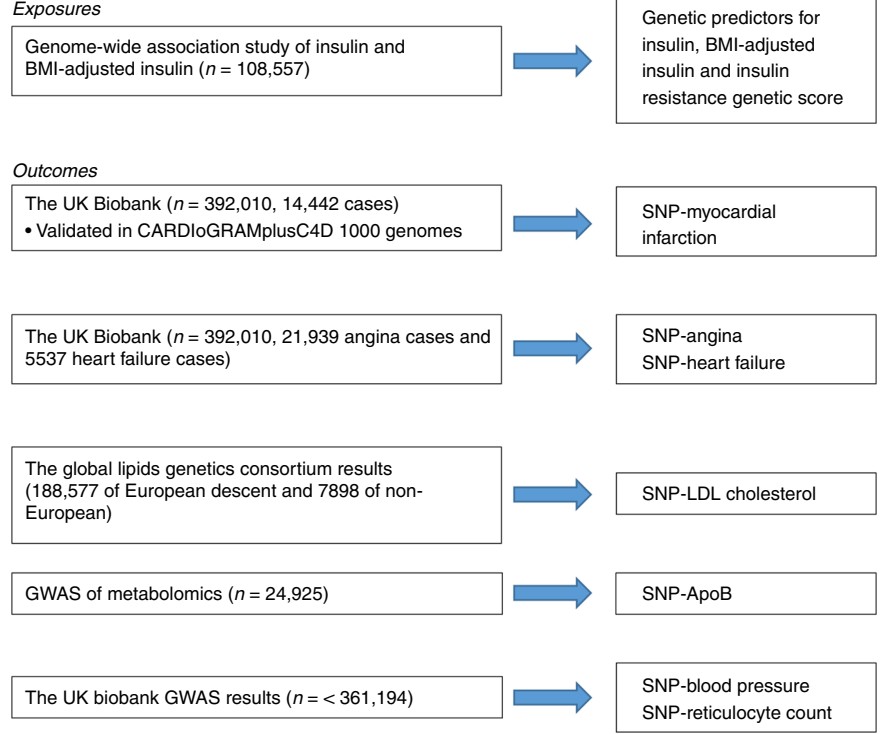

**Fig. 2** Flow chart of data sources

explained by the genetic predictors[31]. Specifically, for cardiovascular events which are the binary outcomes (log OR here), the sample size was calculated based on the effect size (log OR here), the ratio of cases to non-cases in the study, and the variance explained by the genetic predictors[57].

In sensitivity analysis, we conducted sex-specific analysis for all outcomes except LDL cholesterol and ApoB for which sex-specific information is not available. To examine whether effect sizes were larger in men than women, we assessed differences by sex using a $z$-test for the difference in sex-specific estimates (log OR or beta-coefficients) using a one-tailed $p$ value[58].

Given potential bias from invalid instruments when using multiple genetic variants, we also conducted a sensitivity analysis using different statistical methods with different assumptions, i.e., a weighted median[59], a mode-based estimate[60] and Mendelian Randomization Pleiotropy Residual Sum and Outlier (MR-PRESSO) with 100,000 simulations[59]. Specifically, a weighted median is robust to invalid instruments and able to provide consistent estimation even when up to 50% of the weight is from invalid SNPs[59]. The mode-based estimate is based on the assumption that a plurality of genetic variants are valid instruments; i.e., there is no larger subset of invalid instruments estimating the same causal parameter than the subset of valid instruments[60]. MR-PRESSO can detect and as necessary correct for potentially pleiotropic outliers[61]. Given the limited number of SNPs, we did not use MR Egger because it is based on the Instrument Strength Independent of Direct Effect (InSIDE) assumption and thereby is more sensitive to outliers and less efficient than other methods such as the weighted median, the mode-based estimate and MR-PRESSO used here[62].

All statistical analyses were conducted using R version 3.4.4 (R Foundation for Statistical Computing, Vienna, Austria) and the R package "MendelianRandomization"[63].

**Ethical approval**. The UK Biobank has already received ethical approval from the Research Ethics Committee and participants provided written informed consent. The analysis of other publicly available data or summary statistics does not require additional ethical approval.

**Reporting summary**. Further information on research design is available in the Nature Research Reporting Summary linked to this article.

## Data availability
The main outcomes are from the UK Biobank under application (#42468). The data is available from the UK Biobank upon request. Data on coronary artery disease/myocardial infarction have also been contributed by CARDIOGRAMplusC4D investigators and have been downloaded from www.CARDIOGRAMPLUSC4D.ORG. Genetic associations with lipids were obtained from the Global Lipids Genetics

Consortium Results, downloaded from http://csg.sph.umich.edu//abecasis/public/lipids2013/. Genetic associations with apolipoprotein B were obtained from the GWAS of Kettunen et al.[56], downloaded from http://www.computationalmedicine.fi/data#NMR_GWAS. Genetic associations with blood pressure and reticulocyte count were from the UK biobank GWAS results, downloaded from http://www.nealelab.is/uk-biobank/, the results of the GWAS and heritability analyses conducted by the Neale Lab. All the data sources were shown in Fig. 2. The summary data are publicly available.

## Code availability
"ukbmd5", "ukbunpack", "ukbconv" and "ukbgene" were used for data validation, unpacking, format conversion and obtaining genetic data. They are available for the approved UK Biobank application (#42468), in the online system (https://bbams.ndph.ox.ac.uk/ams/). Software code in R for implementing the Mendelian Randomization analysis, is publicly available (https://cran.r-project.org/web/packages/MendelianRandomization/MendelianRandomization.pdf) without restriction[63].

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

## Acknowledgements
The authors would like to thank the UK Biobank for approving our application. The authors would like to thank all investigators of the GWAS applied in this study, for sharing these valuable data. The authors thank Dr. Au Yeung SL for facilitating the storage of UK Biobank genetic data.

## Author contributions
C.M.S. wrote the original draft and J.V.Z. rewrote, J.V.Z. conducted the analysis with the help of C.M.S. and S.L., J.V.Z. drafted the manuscript, C.M.S. and S.L. critically revised it. All authors approved the manuscript.

## Additional information

**Competing interests:** The authors declare no competing interests.

