## [Peer Review File · Communications Biology]

Reviewers' comments:

Reviewer #1 (Remarks to the Author):

Zhao et al perform a large MR study of serum insulin, insulin resistance on myocardial infarction, angina and heart failure using in 392,010 white British from UK biobank. They show genetically predicted insulin was associated with all three outcomes in all individuals, and validate with CAD/MI using Cardiogram data. In sex specific analysis, they show the association is driven in men only but not women, indicating a sex specific effect. Whereas the MR finding of insulin on MI is known, the finding of it being sex-specific is interesting and novel.

Below are some suggestions and comments:

- 1) The authors remove 5 SNVs due to observed genetic associations with other traits, and then perform MR analysis of the remaining 7 SNVs. In sensitivity analysis, the authors perform additional MR tests that account for pleiotropy on these 7 SNVs. The additional MR tests don't necessitate a prior removal of SNVs based on observed pleiotropy therefore it would be good to include all 12 SNVs in the additional MR tests as a sensitivity analysis.
- 2) It would nice to show the actual genetic association results of the tested 7 SNVs in males and females separately, to see the SNV-specific effects.
- 3) Visualization of the association of genetically-predicted insulin and MI, angina and heart failure, can be shown with scatter plots of the effect size (with standard errors) of SNV on insulin vs. effect size of SNV on MI.
- 4) It would be interesting to show if there is an observational correlation between the serum insulin level with MI phenotype in UK biobank in males only, and females only. Although confounding and reverse causation are issues for observational analysis, this can provide support to do MR analysis.
- 5) Please clarify which MR analyses used GWAS summary statistics and which MR tests used individual level genotypes and phenotypes.
- 6) Following point 5, in those instances where you used two sample MR based on GWAS summary statistics, please note that if GWAS summary statistics for both the exposure and outcome were obtained from one sample source, re. UK biobank solely, then the causal estimate will be biased, see PMID 27625185. this should be discussed as a limitation in discussion.
- 7) The sample size for number of cases is much larger in men than women - can you perform power calculations to show specifically that the null results in Women is not due to reduced statistical power?
- 8) It would nice to obtain validation of the sex-specific results in another cohort. If this is not possible, it would be good to mention that additional replication in other cohorts is warranted to provide more support of this finding.

Minor comments

- 9) Table 1: show exact p-value rather than $P=0.000$

Reviewer #2 (Remarks to the Author):

This is a study that makes wonderful use of the Mendelian Randomization method. I found that the study was convincing and that the authors did a fantastic job exploring genetic links between insulin and heart disease by sex using MR. In fact, having reviewed many MR papers in the past, this was one of the more convincing studies that I have reviewed. However, the limitations of the MR method, in particular violations of the exclusion restriction, make MR in general fraught with potentially violated assumptions and other biases. I would like to see the limitations section greatly expanded. Under each of the important limitations that the authors point out, how precisely might violations of assumptions or failure to adequately address limitations bias results? In general, I think this is needed in all MR work, but being concrete about important assumptions and biases builds a much more honest and straightforward scientific paper. I would also like to see the authors reference limitations throughout the paper as they describe results and hypotheses and how they will explore hypotheses, rather than waiting until the end. MR is not in any way a perfect solution to causality in genetics, and so it is important in improving the body of literature in this field that limitations are not simply hastily placed in the discussion of a paper. Other than this important methodological consideration, I believe the authors have done a fantastic job here and, once this issue is addressed, I would recommend publication of this paper.

Reviewer #3 (Remarks to the Author):

This is an article investigating the genetic association between predicted insulin/insulin resistance and myocardial infarction, angina, and heart failure. The authors used Mendelian randomization method in the UK Biobank database. Genetically predicted insulin was associated with myocardial infarction in the overall participants and male subgroup. However, this association was not significant in female subgroup. Regarding angina, predicted insulin had significant association only in male subgroup. This relationship was similar for BMI adjusted insulin level and insulin resistance genetic score. The authors confirm previously known associations of genetically predicted insulin level with myocardial infarction and angina. It is also stated that there is sex specific association, which is only prominent in men. The manuscript is overall well written, and analysis has been done thoroughly by experienced investigators. However, I have the following questions and comments regarding this manuscript.

1. The association between genetically predicted insulin/insulin resistance is already reported in major journals. It is nice to see that this study is replicating previous results. However, it will be more interesting if the authors are able to provide novel insights to this relationship using the one of the largest genetic association databases of UK Biobank.
2. The sex specific effect is interesting. However, there is a large difference in sample size between men and women. Please comment on how this might have affected the results.
3. I wonder if diabetes patients are included in the analysis. In that case, is it possible that diabetes per se, and anti-diabetic medications might have affected the outcomes?
4. Similarly, is there a possibility of reverse causality?
5. I am curious why the authors selected reticulocyte count as one of the outcomes for the genetically predicted insulin level? What is the hypothesis underling this investigation?

Reviewers' comments:

Reviewer #1 (Remarks to the Author):

Zhao et al perform a large MR study of serum insulin, insulin resistance on myocardial infarction, angina and heart failure using in 392,010 white British from UK biobank. They show genetically predicted insulin was associated with all three outcomes in all individuals, and validate with CAD/MI using Cardiogram data. In sex specific analysis, they show the association is driven in men only but not women, indicating a sex specific effect. Whereas the MR finding of insulin on MI is known, the finding of it being sex-specific is interesting and novel.

Thank you very much for your positive comment.

Below are some suggestions and comments:

1) The authors remove 5 SNVs due to observed genetic associations with other traits, and then perform MR analysis of the remaining 7 SNVs. In sensitivity analysis, the authors perform additional MR tests that account for pleiotropy on these 7 SNVs. The additional MR tests don't necessitate a prior removal of SNVs based on observed pleiotropy therefore it would be good to include all 12 SNVs in the additional MR tests as a sensitivity analysis.

Thank you very much for your comment. Please accept our apologies for being unclear. We did include a sensitivity analysis using all the 12 genetic variants for insulin (Supplemental Table 5 as shown below). We have amended the title to be clearer and more explicit.

From: "Sensitivity analyses on the associations of genetically predicted insulin with myocardial infarction, angina and heart failure with potentially pleiotropic SNPs"

To: "Sensitivity analyses showing the associations of genetically predicted insulin with myocardial infarction, angina and heart failure including all potentially pleiotropic SNPs".

Supplemental Table 5. Sensitivity analyses showing the associations of genetically predicted insulin with myocardial infarction, angina and heart failure including all potentially pleiotropic SNPs

Outcomes	Sex	Insulin			Insulin adjusted for BMI		
		#SNPs	OR	95% CI	#SNPs	OR	95% CI
Myocardial infarction	Overall	12	1.28	0.58 to 2.82	12	2.75	1.68 to 4.53
	Men	12	1.83	0.80 to 4.21	12	3.59	1.67 to 7.71
	Women	12	0.40	0.14 to 1.14	12	0.99	0.31 to 3.14
Angina	Overall	12	1.41	0.72 to 2.77	12	1.89	1.04 to 3.41
	Men	12	1.65	0.82 to 3.32	12	2.10	0.98 to 4.50
	Women	12	1.07	0.48 to 2.41	12	1.55	0.75 to 3.17
Heart failure	Overall	12	1.22	0.60 to 2.46	12	1.48	0.60 to 3.67
	Men	12	1.61	0.70 to 3.72	12	2.16	0.86 to 5.44
	Women	12	0.62	0.15 to 2.53	12	0.58	0.08 to 4.24

CI, confidence interval; OR, odds ratio; SNP, single nucleotide polymorphism

2) It would nice to show the actual genetic association results of the tested 7 SNVs in males and females separately, to see the SNV-specific effects.

Thank you very much for your comment. We have added the results in Supplemental Table 2 as follows:

Supplemental Table 2. Sex-specific associations of each genetic variant for insulin with myocardial infarction, angina and heart failure

SNP	Effect allele	MI in men		MI in women		Angina in men		Angina in women		Heart failure in men		Heart failure in women	
		beta	SE	beta	SE	beta	SE	beta	SE	beta	SE	beta	SE
rs1530559	A	0.01	0.01	-0.02	0.03	0.01	0.01	-0.003	0.02	0.04	0.02	0.02	0.04
rs2745353	T	0.02	0.01	0.04	0.03	0.02	0.01	0.03	0.02	-0.01	0.02	-0.02	0.04
rs2820436	C	0.02	0.01	-0.03	0.03	0.02	0.01	0.001	0.02	0.02	0.02	-0.09	0.04
rs2972143	G	0.06	0.01	-0.01	0.03	0.04	0.01	0.02	0.02	0.03	0.02	-0.06	0.04
rs4865796	A	0.03	0.02	0.01	0.03	0.01	0.01	-0.02	0.02	-0.01	0.02	0.06	0.04
rs731839	G	0.03	0.01	-0.001	0.03	0.03	0.01	0.01	0.02	-0.03	0.02	-0.001	0.04
rs983309	T	-0.002	0.02	-0.01	0.04	-0.01	0.02	-0.04	0.03	-0.03	0.04	0.04	0.05

MI, myocardial infarction; SE, standard error

In results, paragraph 2, we added “7 SNPs were used (Supplemental Table 1 and Supplemental Table 2)”.

3) Visualization of the association of genetically-predicted insulin and MI, angina and heart failure, can be shown with scatter plots of the effect size (with standard errors) of SNV on insulin vs. effect size of SNV on MI.

Thank you very much for your comment. We have added the scatter plot in Supplemental Figure 1, as follows:

Supplemental Figure 1. Scatter plot for genetically predicted insulin and myocardial infarction, angina and heart failure

*In results, paragraph 5, we added “**Genetically predicted insulin, BMI-adjusted insulin and insulin resistance score were all positively associated with MI overall (Table 1 and Supplemental Figure 1)**”.*

4) It would be interesting to show if there is an observational correlation between the serum insulin level with MI phenotype in UK biobank in males only, and females only. Although confounding and reverse causation are issues for observational analysis, this can provide support to do MR analysis.

Thank you very much for your helpful comment. It is a great idea and would be very interesting to compare the sex-specific associations in MR with those in conventional observational studies. However, serum insulin is not currently available in the UK Biobank, so we cannot do this analysis at the moment. We really look forward to doing this analysis once the data is available in the future.

5) Please clarify which MR analyses used GWAS summary statistics and which MR tests used individual level genotypes and phenotypes.

Thank you very much for your comment. We have added clarification on the use of summary statistics and individual level data. The revisions are as follows:

(Methods-Genetic associations with MI, angina and heart failure)

“Genetic associations with MI, angina and heart failure were obtained using individual-level data in the UK Biobank (under the application #42468), with validation for MI using summary statistics from CARDIoGRAMplusC4D 1000 Genomes¹.”

(Methods-Genetic associations with LDL-cholesterol and ApoB)

“Genetic associations with LDL-cholesterol (as inverse normal transformed effect sizes), adjusted for age, age² and sex, were obtained from the Global Lipids Genetics Consortium Results summary statistics ...”

(Methods-Genetic associations with blood pressure and reticulocyte count)

“We obtained overall and sex-specific genetic associations with blood pressure and reticulocyte count using summary statistics from the UK Biobank, provided by Neale Lab (<http://www.nealelab.is/uk-biobank/>)...”

6) Following point 5, in those instances where you used two sample MR based on GWAS summary statistics, please note that if GWAS summary statistics for both the exposure and outcome were obtained from one sample source, re. UK biobank solely, then the causal estimate will be biased, see PMID 27625185. this should be discussed as a limitation in discussion.

Thank you very much for your comment. We agree the causal estimates could be biased if the genetic associations with the exposure and outcome are from the same sample. However, we do not think this is the case in this study. Specifically, genetic associations with insulin and insulin resistance were from the Meta-Analyses of Glucose and Insulin-related traits Consortium (MAGIC), whilst the genetic associations with MI, angina, heart failure, blood pressure and reticulocyte count were from the UK Biobank, and genetic associations with LDL-cholesterol were obtained from the Global Lipids Genetics Consortium Results. We have expanded the discussion to explain the difference in bias for one-sample and two-sample Mendelian randomization (MR), as follows:

(Discussion-paragraph 3)

From:

“In addition, the sample for genetic variants on insulin has no overlap with the UK Biobank. As such, any relationship of the genetic variants to unmeasured confounders is not expected to exist coincidentally in the samples for insulin or insulin resistance and for the outcomes, due to the different data structures².”

To:

“In addition, the sample for genetic variants on insulin has no overlap with the UK Biobank. Two-sample MR is less biased than one-sample MR³, because any relation of the genetic variants with unmeasured confounders is not expected to exist coincidentally in both

the sample providing genetic associations with insulin or insulin resistance and the sample providing genetic associations with the outcomes, due to the different data structures². If bias did occur due to weak instruments, it is often towards the null, whereas in one-sample MR the bias is towards the direction of the conventional observational studies³.

7) The sample size for number of cases is much larger in men than women - can you perform power calculations to show specifically that the null results in Women is not due to reduced statistical power?

Thank you very much for your comment. We have added power calculation in the methods and results as follows:

*In the methods, we added “**Power calculations were conducted overall and by sex. MR studies require larger sample sizes than conventional observational studies, because the sample size needed for MR is the sample size for the conventional observational study divided by the variance in the exposure explained by the genetic predictors⁴.**”*

In the results, we added

“The replication for MI using a different study provides additional validation, and enabled us to test causality in a cost-efficient way⁵. Specifically, the studies for MI with over 56,000 cases, at an approximate R^2 of 0.01 (variance in insulin/BMI-adjusted insulin explained by the genetic predictors), has 0.8 power to detect an odds ratio (OR) of about 1.14 per one standard deviation increase in the exposure. The UK biobank has 0.8 power to detect an OR of about 1.22 for MI overall, of 1.28 for MI in men and 1.43 in women; an OR of 1.20 for angina overall, of 1.25 for angina in men and 1.33 in women; an OR of 1.40 for heart failure overall, of 1.47 for heart failure in men and 1.60 in women⁴. The larger number of cases in men than women enabled us to test a smaller effect size in men, however, there is sufficient power for both men and women, especially when using the insulin resistance score. The difference in power does not explain the sex-disparity in the magnitude of the point estimates and/or direction of associations.”

8) It would nice to obtain validation of the sex-specific results in another cohort. If this is not possible, it would be good to mention that additional replication in other cohorts is warranted to provide more support of this finding.

*Thank you very much for your comment. It would be great to replicate in another cohort, however, we cannot find another cohort which can provide sex-specific genetic associations. We have added in the limitations “**Validation of the sex-specific associations in another cohort is warranted.**”*

*We have also added in the conclusions “**Replication in other cohorts is needed.**”*

Minor comments

9) Table 1: show exact p-value rather than $P=0.000$

Thank you very much for your comment. We have replaced the p-values with the exact values as follows:

Table 1. Associations of genetically predicted insulin, insulin adjusted for BMI and insulin resistance genetic score with myocardial infarction, angina, and heart failure

Outcomes	Data source	Sex	#cases	Insulin			Insulin adjusted for BMI			Insulin resistance genetic score		
				OR	95% CI	p	OR	95% CI	p	OR	95% CI	p
Myocardial infarction	UK Biobank	Overall	14,442	2.87	1.30 to 6.33	0.009	2.63	1.36 to 5.07	0.004	3.22	1.86 to 5.58	2.4×10⁻⁵
		Men	11,182	4.27	1.60 to 11.3	0.004	3.60	1.46 to 8.83	0.005	4.18	2.23 to 7.82	6.9×10⁻⁶
		Women	3,260	0.80	0.23 to 2.84	0.73	0.96	0.30 to 3.09	0.95	1.40	0.46 to 4.29	0.55
	CARDIoGRAMplusC4D 1000 Genomes	Overall	42,561	1.90	1.04 to 3.49	0.04	2.20	1.08 to 4.50	0.03	--	--	--
Angina	Meta-analysis	Overall	57,003	2.21	1.37 to 3.58	0.001	2.42	1.49 to 3.93	0.0003	--	--	--
Heart failure	UK Biobank	Overall	21,939	2.05	0.87 to 4.83	0.10	1.90	0.92 to 3.94	0.08	2.87	1.83 to 4.50	3.7×10⁻⁶
		Men	14,331	2.93	1.27 to 6.73	0.01	2.74	1.21 to 6.19	0.02	3.56	2.06 to 6.16	8.3×10⁻⁶
		Women	7,608	1.10	0.38 to 3.18	0.87	1.00	0.46 to 2.19	1.00	1.95	0.93 to 4.12	0.08
Heart failure	UK Biobank	Overall	5,537	0.98	0.37 to 2.61	0.97	1.18	0.45 to 3.14	0.74	1.00	0.42 to 2.34	1.00
		Men	3,935	1.17	0.30 to 4.64	0.82	1.90	0.62 to 5.82	0.26	1.72	0.62 to 4.76	0.31
		Women	1,602	0.64	0.06 to 7.18	0.72	0.37	0.03 to 4.05	0.41	0.28	0.06 to 1.33	0.11

BMI, body mass index; CI, confidence interval; OR, odds ratio

Inverse variance weighting with random effects was used for insulin and insulin adjusted for BMI; logistic regression was used for insulin resistance genetic score.

Reviewer #2 (Remarks to the Author):

This is a study that makes wonderful use of the Mendelian Randomization method. I found that the study was convincing and that the authors did a fantastic job exploring genetic links between insulin and heart disease by sex using MR. In fact, having reviewed many MR papers in the past, this was one of the more convincing studies that I have reviewed. However, the limitations of the MR method, in particular violations of the exclusion restriction, make MR in general fraught with potentially violated assumptions and other biases. I would like to see the limitations section greatly expanded. Under each of the important limitations that the authors point out, how precisely might violations of assumptions or failure to adequately address limitations bias results? In general, I think this is needed in all MR work, but being concrete about important assumptions and biases builds a much more honest and straightforward scientific paper. I would also like to see the authors reference limitations throughout the paper as they describe results and hypotheses and how they will explore hypotheses, rather than waiting until the end. MR is not in any way a perfect solution to causality in genetics, and so it is important in improving the body of literature in this field that limitations are not simply hastily placed in the discussion of a paper. Other than this important methodological consideration, I believe the authors have done a fantastic job here and, once this issue is addressed, I would recommend publication of this paper.

Thank you very much indeed for the positive comments. We have expanded the limitations section greatly, to address in detail and more precisely of the limitations. As you suggested, we re-arranged the discussion, and put some of the limitations concerning power calculation to the methods and results. The re-arrangement has been shown with track changes throughout the paper. The discussion has been expanded as follows:

From:

“First, MR is based on three assumptions, i.e., relevance, independence and exclusions-restriction (no pleiotropy). We used genetic variants strongly associated with insulin and insulin resistance identified in large GWAS^{6,7}, as previously^{8,9}. We checked for associations with potential confounders, such as socioeconomic position and lifestyle in the UK Biobank. In addition, the sample for genetic variants on insulin has no overlap with the UK Biobank. As such, any relationship of the genetic variants to unmeasured confounders is not expected to exist coincidentally in the samples for insulin or insulin resistance and for the outcomes, due to the different data structures².... To detect known potential pleiotropy we checked in three comprehensive curated databases.”

To:

“First, MR is based on three assumptions, i.e., the genetic variants are strongly related to the exposure, are not related to the exposure-outcome confounders, and the genetic variants are related to the outcomes only via influencing the exposure^{10,11}. To satisfy the first assumption, we used genetic variants strongly associated with insulin and insulin resistance identified in large GWAS^{6,7}, as previously^{8,9}. To satisfy the second assumption,

we checked for associations with known exposure-outcome confounders, including socioeconomic position and lifestyle in the UK Biobank, where there was no association with these potential confounders. In addition, the sample for genetic variants on insulin has no overlap with the UK Biobank. Two-sample MR is less biased than one-sample MR³, because any relation of the genetic variants with unmeasured confounders is not expected to exist coincidentally in both the sample providing genetic associations with insulin or insulin resistance and the sample providing genetic associations with the outcomes, due to the different data structures². If bias did occur due to weak instruments, it is often towards the null, whereas in one-sample MR the bias is towards the directions of the conventional observational studies³. ...To test the assumption of pleiotropy, we checked for the known potential pleiotropy in three comprehensive curated databases.”

From:

“Fourth, our study could be affected by survivor bias (selection bias)¹², and by competing risk for specific causes of death that share risk factors.”

To:

“Fourth, our study could be affected by survivor bias (selection bias)¹², and by competing risk for specific causes of death that share risk factors. Specifically, the estimates for a potentially harmful exposure might be biased towards being less harmful if people with higher levels of exposures were already dead and not selected into the study, as in the obesity paradox¹³.”

Reviewer #3 (Remarks to the Author):

This is an article investigating the genetic association between predicted insulin/insulin resistance and myocardial infarction, angina, and heart failure. The authors used Mendelian randomization method in the UK Biobank database. Genetically predicted insulin was associated with myocardial infarction in the overall participants and male subgroup. However, this association was not significant in female subgroup. Regarding angina, predicted insulin had significant association only in male subgroup. This relationship was similar for BMI adjusted insulin level and insulin resistance genetic score. The authors confirm previously known associations of genetically predicted insulin level with myocardial infarction and angina. It is also stated that there is sex specific association, which is only prominent in men. The manuscript is overall well written, and analysis has been done thoroughly by experienced investigators. However, I have the following questions and comments regarding this manuscript.

Thank you very much for your positive comment.

1. The association between genetically predicted insulin/insulin resistance is already reported in major journals. It is nice to see that this study is replicating previous results. However, it will be more interesting if the authors are able to provide novel insights to this relationship using the one of the largest genetic association databases of UK Biobank.

Thank you very much for your comment. Our study is consistent with previous study on genetically predicted insulin and ischemic heart disease (IHD). Our study adds to the current evidence by showing the sex-specific associations of genetically predicted insulin and insulin resistance in subtypes of IHD, suggesting a sex-disparity in these associations. We have expanded the discussion as follows:

(Discussion, paragraph 4)

“Our study, together with previous evidence^{14,15}, suggests that insulin and insulin resistance have symbiotic roles that may both ultimately play a role in CVD. Our study adds to the current evidence by showing a sex-disparity in these associations.”

2. The sex specific effect is interesting. However, there is a large difference in sample size between men and women. Please comment on how this might have affected the results.

Thank you very much for your comment. We agree there is a larger sample size in men than in women, however, the difference in sample size should only affect the precision of the estimates, rather than the magnitude of the point estimates or the direction of the associations. We have added power calculation in the methods and results as follows:

In the methods, we added **“Power calculations were conducted overall and by sex. MR studies require larger sample sizes than conventional observational studies, because the sample size needed for MR is the sample size for the conventional observational study divided by the variance in the exposure explained by the genetic predictors⁴.”**

In the results, we added

“The replication for MI using a different study provides additional validation, and enabled us to test causality in a cost-efficient way⁵. Specifically, the studies for MI with over 56,000 cases, at an approximate R^2 of 0.01 (variance in insulin/BMI-adjusted insulin explained by the genetic predictors), has 0.8 power to detect an odds ratio (OR) of about 1.14 per one standard deviation increase in the exposure. The UK biobank has 0.8 power to detect an OR of about 1.22 for MI overall, of 1.28 for MI in men and 1.43 in women; an OR of 1.20 for angina overall, of 1.25 for angina in men and 1.33 in women; an OR of 1.40 for heart failure overall, of 1.47 for heart failure in men and 1.60 in women⁴. The larger number of cases in men than women enabled us to test a smaller effect size in men, however, there is sufficient power for both men and women, especially when using the insulin resistance score. The difference in power does not explain the sex-disparity in the magnitude of the point estimates and/or direction of associations.”

3. I wonder if diabetes patients are included in the analysis. In that case, is it possible that diabetes per se, and anti-diabetic medications might have affected the outcomes?

Thank you very much for your comment. We did not specifically exclude people with type 2 diabetes from the analysis. Diabetes or anti-diabetic medications might affect the outcomes but should not affect the genetic predictors for insulin or insulin resistance, so the associations of genetically predicted insulin or insulin resistance should not be confounded by diabetes or anti-diabetic medications. It is possible that adjusting for diabetes or anti-diabetic medications might improve the precision of the estimates. However, it is also possible that diabetes is a mediator of the association of insulin or insulin-resistance with the outcomes, in which case adjusting for diabetes (by adjustment or exclusion) would give the direct effect instead of the total effect and thereby introduce a bias. As such, we prefer not to adjust for or exclude by diabetes or diabetes medication status. We have expanded the discussion to explain this point as follows:

“Seventh, some of the participants may have comorbidities such as type 2 diabetes and may be taking medications for these comorbidities. Co-morbidities and their treatment may affect the cardiovascular outcomes, but should not affect the genetic predictors of exposures, so they are not confounders but their inclusion could improve the precision of the estimates. However, co-morbidities could also be consequences of insulin and insulin resistance so their consideration in the model would give the direct effects of insulin rather than the total effect sought, i.e., might create bias. As such, we did not account for co-morbidities or their treatment by adjustment or restriction, so as to obtain an unbiased, though possibly less precise, estimates.”

4. Similarly, is there a possibility of reverse causality?

Thank you very much for your comment. Reverse causality, i.e., cardiovascular events leading to abnormal insulin or insulin resistance, is not a major concern in this study. People with cardiovascular events may change their lifestyle, which may be beneficial for lowering insulin resistance, however, this cannot explain the positive associations of insulin or insulin resistance with cardiovascular events in this study, because it cannot change the genetic predictors. Moreover, all SNPs are genome-wide significant SNPs for insulin or insulin resistance, none of them are genome-wide significant for myocardial infarction (MI), angina or heart failure. We have expanded the discussion as follows:

“Eighth, reverse causality may occur if people with cardiovascular events change their lifestyle thereby affecting insulin or insulin resistance. However, these changes would not affect genetically predicted insulin or insulin resistance. None of the genetic variants are genome-wide significant for cardiovascular events, so it is unlikely that they predict insulin or insulin resistance by affecting cardiovascular events.”

5. I am curious why the authors selected reticulocyte count as one of the outcomes for the genetically predicted insulin level? What is the hypothesis underling this investigation?

Thank you very much for your comments. Red blood cell traits have long been suspected to play a role in cardiovascular disease¹⁶⁻¹⁸, although it is not clear which specific trait is causal. The most recent evidence from an MR study published in Cell suggests reticulocytes are related to higher cardiovascular risk¹⁹, although more validation is needed. Based on the best evidence available, we used reticulocyte count as one of the outcomes. We have added further explanation on this point as follows:

(Introduction-paragraph 3)

From:

“Here, we used MR to assess overall and sex-specific effects of insulin, and for completeness insulin resistance, on MI, angina, heart failure and their key risk factors (low density lipoprotein (LDL) cholesterol, apolipoprotein B (ApoB)²⁰, blood pressure and reticulocyte count, a recently identified causal factor for CVD¹⁹) using individual data in a large cohort, the UK Biobank²¹, or the largest available genome wide association study (GWAS).”

To:

“Here, we used MR to assess overall and sex-specific effects of insulin, and for completeness insulin resistance, on MI, angina, heart failure and their key risk factors (low density lipoprotein (LDL) cholesterol, apolipoprotein B (ApoB)²⁰, and blood pressure) using individual data in a large cohort, the UK Biobank²¹, or the largest available genome wide association study (GWAS). Red blood cell attributes have long been suspected to be relevant to cardiovascular disease¹⁸, however, which trait matters is not well-established.”

The most recent evidence from an MR study suggests the red blood cell trait, reticulocyte count, may be a causal factor for IHD¹⁹, so we similarly examined the role of insulin and insulin resistance in reticulocyte count.”

References

- 1 Nikpay, M. *et al.* A comprehensive 1,000 Genomes-based genome-wide association meta-analysis of coronary artery disease. *Nature Genet* **47**, 1121-1130 (2015).
- 2 Tchetgen Tchetgen, E. J., Walter, S. & Glymour, M. M. Commentary: building an evidence base for mendelian randomization studies: assessing the validity and strength of proposed genetic instrumental variables. *Int J Epidemiol* **42**, 328-331 (2013).
- 3 Burgess, S., Davies, N. M. & Thompson, S. G. Bias due to participant overlap in two-sample Mendelian randomization. *Genet. Epidemiol.* **40**, 597-608 (2016).
- 4 Freeman, G., Cowling, B. J. & Schooling, C. M. Power and sample size calculations for Mendelian randomization studies using one genetic instrument. *Int J Epidemiol* **42**, 1157-1163 (2013).
- 5 Burgess, S. *et al.* Using published data in Mendelian randomization: a blueprint for efficient identification of causal risk factors. *Eur J Epidemiol* **30**, 543-552 (2015).
- 6 Scott, R. A. *et al.* Large-scale association analyses identify new loci influencing glycemic traits and provide insight into the underlying biological pathways. *Nature Genet* **44**, 991-1005 (2012).
- 7 Scott, R. A. *et al.* Common genetic variants highlight the role of insulin resistance and body fat distribution in type 2 diabetes, independent of obesity. *Diabetes* **63**, 4378-4387 (2014).
- 8 Tikkanen, E. *et al.* Genetic support for the causal role of insulin in coronary heart disease. *Diabetologia* **59**, 2369-2377 (2016).
- 9 Zhan, Y. *et al.* Exploring the Causal Pathway From Telomere Length to Coronary Heart Disease: A Network Mendelian Randomization Study. *Circ. Res.* **121**, 214-219 (2017).
- 10 Davies, N. M., Holmes, M. V. & Davey Smith, G. Reading Mendelian randomisation studies: a guide, glossary, and checklist for clinicians. *BMJ* **362**, k601 (2018).
- 11 Pearl, J. *Causality*. Cambridge University Press, 2000.
- 12 Schooling, C. M. Selection bias in population-representative studies? A commentary on Deaton and Cartwright. *Soc. Sci. Med.* **210**, 70 (2018).
- 13 Schooling, C. M., Cowling, B. J. & Jones, H. E. Selection bias in cohorts of cases. *Prev. Med.* **57**, 247-248 (2013).
- 14 Czech, M. P. Insulin action and resistance in obesity and type 2 diabetes. *Nat. Med.* **23**, 804-814 (2017).
- 15 Astley, C. M. *et al.* Genetic Evidence That Carbohydrate-Stimulated Insulin Secretion Leads to Obesity. *Clin. Chem.* **64**, 192-200 (2018).
- 16 Zhao, J. V. & Schooling, C. M. Risk for Arterial and Venous Thrombosis in Patients With Myeloproliferative Neoplasms. *Ann. Intern. Med.* **169**, 267-268 (2018).
- 17 Hultcrantz, M. *et al.* Risk for Arterial and Venous Thrombosis in Patients With Myeloproliferative Neoplasms: A Population-Based Cohort Study. *Ann. Intern. Med.* **168**, 317-325 (2018).
- 18 Burch, G. Erythrocytosis and ischemic myocardial disease. *Am. Heart J.* **62**, 139-140 (1961).
- 19 Astle, W. J. *et al.* The Allelic Landscape of Human Blood Cell Trait Variation and Links to Common Complex Disease. *Cell* **167**, 1415-1429 (2016).

- 20 Ference, B. A. *et al.* Association of Triglyceride-Lowering LPL Variants and LDL-C-Lowering LDLR Variants With Risk of Coronary Heart Disease. *JAMA* **321**, 364-373 (2019).
- 21 Suldwon, C. *et al.* UK Biobank: An Open Access Resource for Identifying the Causes of a Wide Range of Complex Diseases of Middle and Old Age. *PLoS Med* **12**, e1001779 (2015).

REVIEWERS' COMMENTS:

Reviewer #1 (Remarks to the Author):

The authors have adequately responded to all of my comments. One minor comment is please provide details of the method used for the power calculations.

Reviewer #2 (Remarks to the Author):

The reviewers have responded to my comments thoughtfully and carefully. I recommend this paper for publication.

Reviewer #3 (Remarks to the Author):

I have no further comments.

Reviewers' comments:

Reviewer #1 (Remarks to the Author):

The authors have adequately responded to all of my comments. One minor comment is please provide details of the method used for the power calculations.

Thank you very much for the positive comments. We have added more details on the power calculation, as follows:

(Methods, paragraph 9)

“Specifically, for cardiovascular events which are the binary outcomes, the sample size was calculated based on the effect size (log odds ratio here), the ratio of cases to non-cases in the study, and the variance explained by the genetic predictors¹.”

Reviewer #2 (Remarks to the Author):

The reviewers have responded to my comments thoughtfully and carefully. I recommend this paper for publication.

Thank you very much for the positive comments.

Reviewer #3 (Remarks to the Author):

I have no further comments.

Thank you very much for the positive comments.

Reference:

- 1 Burgess, S. Sample size and power calculations in Mendelian randomization with a single instrumental variable and a binary outcome. *Int. J. Epidemiol.* **43**, 922-929 (2014).